# Analyses of the Association between Thyroid Cancer and Osteoporosis/Fracture Histories: A Cross-Sectional Study Using KoGES HEXA Data

**DOI:** 10.3390/ijerph18094732

**Published:** 2021-04-29

**Authors:** Young-Ju Jin, Chang-Myeon Song, Bum-Jung Park, Hyo-Geun Choi

**Affiliations:** 1Department of Otorhinolaryngology-Head and Neck Surgery, Wonkwang University Hospital, Wonkwang University College of Medicine, Iksan 54538, Korea; chindol1@wku.ac.kr; 2Department of Otolaryngology-Head and Neck Surgery, Hanyang University College of Medicine, Seoul 04763, Korea; cmsong@hanyang.ac.kr; 3Department of Otorhinolaryngology-Head and Neck Surgery, Hallym University College of Medicine, Anyang 14103, Korea; pbj426@hallym.ac.kr; 4Hallym Data Science Laboratory, Hallym University College of Medicine, Anyang 14103, Korea

**Keywords:** thyroid cancer, osteoporosis, fracture, cross-sectional study

## Abstract

(1) Background: The purpose of this study was to determine the association among thyroid cancer, osteoporosis and fracture history. (2) Methods: The data collected from 2004 through 2016 for the Korean Genome and Epidemiology Study were retrieved. For a total of 1349 participants with thyroid cancer and 163,629 control participants, the odds ratios (ORs) with 95% confidence intervals (CIs) of osteoporosis and fracture history were evaluated using a logistic regression model. (3) Results: The adjusted ORs of osteoporosis comparing thyroid cancer with the control group were 1.41 (95% CI = 1.18–1.70, *p* < 0.001) for all participants and 1.43 (95% CI = 1.19–1.71, *p* < 0.001) for women. The adjusted ORs of fracture history comparing these two groups were not significantly associated within the entire group of participants, men only or women only. (4) Conclusions: The adjusted OR of osteoporosis was significantly higher than 1, comparing thyroid cancer with the control group, especially in women. The adjusted OR of fractures was not significantly higher than 1, comparing the thyroid cancer group with the control group.

## 1. Introduction

Osteoporosis is a bone disease that is characterized by the loss of bone mass and strength, which lead to fragility fractures and thus severe morbidities and mortality. The prevalence of osteoporosis and osteopenia in the US is 35.5 million in women and 18.2 million in men [1]. Moreover, approximately 50% of females and 20% of males experience fragility fractures of the hip, wrist or spine in their aged life [2]. In Korea, the prevalence of osteoporosis was reported to be 38.0% in females and 7.3% in males aged over 50 years old [3]. The main causes of osteoporosis are related to age, calcium deficiency, vitamin D deficiency, secondary hyperparathyroidism, inflammatory diseases such as rheumatoid arthritis, glucocorticoid use, excessive alcohol use and estrogen deficiency, especially in postmenopausal women [4,5].

Thyroid cancer is the most common endocrine malignancy. The incidence of thyroid cancer increased 3.6% per year during 1974–2013 in a US study [6]. The incidence of thyroid cancer increased 22.6% during 1999–2011 in Korea [7]. Regarding treatment, various kinds of thyroidectomy, including partial, subtotal or total thyroidectomy, are performed and followed with radioactive iodine (RAI) therapy after thyrotropin (TSH) suppression.

Previous studies have suggested that the incidence of osteoporosis is higher in patients with than in those without thyroid cancer [8,9,10,11]. In a population-based observational cohort study in Taiwan, the total thyroidectomy group showed a significantly elevated long-term risk of osteoporosis [12]. Although osteoporosis causes fragility fractures, the relationship between thyroid cancer and fracture history remains unclear. Some studies have suggested that the risk of fracture is significantly increased in thyroid cancer patients [13,14]. However, other studies have suggested that the risk of fracture is not higher in thyroid cancer or total thyroidectomy patients than in patients who have not undergone thyroidectomy [9,12]. Moreover, patients with TSH suppression therapy but not those with thyroid cancer experience a high number of fractures [15,16,17]. Therefore, we compared the prevalence of fractures as well as osteoporosis between thyroid cancer patients and controls.

The aim of this study was to evaluate the association among thyroid cancer, osteoporosis and fractures based on Korean Genome and Epidemiology Study (KoGES) health examinee (HEXA) data. The odds ratio itself compares these two groups.

## 2. Materials and Methods

### 2.1. Study Population and Data Collection

The ethics committee of Hallym University (20 February 2019) approved the use of these data. The requirement for written informed consent was waived by the institutional review board. This prospective cohort study used data collected from 2004 through 2016 for the Korean Genome and Epidemiology Study (KoGES). A detailed description of these data is provided in a previous study [18]. Among the KoGES consortium data, we used KoGES health examinee (HEXA) data for urban residence participants ≥ 40 years old.

### 2.2. Participants Selection

Among 173,209 participants, we excluded participants who lacked data on height or weight (*n* = 934); smoking history (*n* = 1390); alcohol consumption habits (*n* = 331); hypertension, diabetes mellitus, or dyslipidemia histories (*n* = 203); arthritis or thyroid disease histories (*n* = 60); and osteoporosis history (*n* = 37). Among the thyroid cancer participants, we excluded those who previously had other types of cancer (gastric cancer, hepatic cancer, colon cancer, breast cancer, lung cancer, uterine cervix cancer, or bladder cancer, *n* = 83). Among the control participants without a history of thyroid cancer, participants with a history of other cancers were excluded (*n* = 5193). Finally, 1349 thyroid cancer and 163,629 control (no history of thyroid cancer) participants were selected (Figure 1). Then, we analyzed the histories of osteoporosis between the thyroid cancer and control participants (primary objective).

Then, we analyzed the histories of any fractures between the thyroid cancer and control participants (secondary objective). In this study, we excluded 64 participants with thyroid cancer and 35,957 control participants who did not have data on their fracture history.

### 2.3. Survey

The participants were asked whether they previously had cancer, including thyroid cancer, gastric cancer, hepatic cancer, colon cancer, breast cancer, lung cancer, cervical cancer, or bladder cancer, by trained interviewers. They were also asked about their past medical history regarding hypertension, diabetes mellitus, dyslipidemia, arthritis, and thyroid diseases (hyperthyroidism or hypothyroidism). Body mass index (BMI) was calculated in kg/m^2^ using the health checkup data. The patients were categorized as nonsmokers (smoked <100 cigarettes in their entire life), past smokers (quit more than one year ago), or current smokers. According to their alcohol consumption habits, the patients were categorized as non-drinkers (no history of alcohol consumption in their entire life), past drinkers (quit alcohol consumption), or current drinkers. The participants were asked whether they had been diagnosed with osteoporosis or any fractures.

### 2.4. Statistical Analyses

The chi-square test was used to compare the rates of each sex, hypertension, diabetes mellitus, dyslipidemia, arthritis, any thyroid disease, smoking, alcohol consumption, and osteoporosis. Independent *t*-tests were used to compare age and BMI.

To analyze the OR of thyroid cancer for osteoporosis, a logistic regression model was used. For the primary objective, crude and adjusted models (age, sex, BMI, hypertension, diabetes mellitus, dyslipidemia, arthritis, thyroid disease histories, smoking, and alcohol consumption) were used. For the secondary objective, crude and adjusted models (previous adjusted model plus osteoporosis histories) were used.

Two-tailed analyses were conducted, and *p* values less than 0.05 were considered to indicate significance. The results were statistically analyzed using SPSS, version 24.0 (IBM, Armonk, NY, USA).

## 3. Results

### 3.1. Detailed Descriptions

#### 3.1.1. General Characteristics of Participants

The proportions of women (91.5% vs. 65.4%), patients with thyroid disease (18.8% vs. 4.5%), non-smokers (92.7% vs. 72.7%), non-drinkers (66.8% vs. 50.4%) and patients with osteoporosis (11.0% vs. 6.7%) were higher in the thyroid cancer group than in the control group with statistical significance. On the other hand, the proportions of past smokers (4.5% vs. 14.7%), current smokers (2.7% vs. 12.7%), past drinkers (3.0% vs. 3.8%) and current drinkers (30.2% vs. 45.8%) were lower in the thyroid cancer group than in the control group. The mean age was lower (52.3 years vs. 53.0 years) in the thyroid cancer group compared with the control group with statistical significance (Table 1).

#### 3.1.2. ORs of Osteoporosis Comparing Participants with Thyroid Cancer to Control

The adjusted ORs of osteoporosis comparing participants with thyroid cancer to control were 1.41 (95% CI = 1.18–1.70, *p* <0.001) in all participants and 1.43 (95% CI = 1.19–1.71, *p* < 0.001) in women (Table 2). On the other hand, a statistical difference was not present between the control group and men.

#### 3.1.3. ORs of Fracture Histories Comparing Participants with Thyroid Cancer to Control

The adjusted ORs of thyroid cancer for fracture history were not statistically significant or larger than 1 in the entire group of participants, men only or women only (Table 3).

## 4. Discussion

In our study, the ORs of osteoporosis were significantly higher than 1, comparing participants with thyroid cancer with control, especially in women over 40 years old. Fracture history did not significantly differ between the thyroid cancer group and the control group.

In accordance with the results of our study, in previous studies, the risk of osteoporosis was significantly higher in the thyroid cancer group than in the control group. In a large case-control study in the US (*n* = 10,370), osteoporosis was more common in the thyroid cancer group than in the control group (7.3% vs. 5.3%) [9]. In a population-based US study (*n* = 3706), the HRs of osteoporosis were higher 1–5 years after thyroid cancer diagnoses and 5–10 years after thyroid cancer diagnoses in both the age groups <40 years and >40 years than in the control group [8]. In a small Japanese case-control study (*n* = 35), the prevalence of severe osteoporosis was significantly higher in the papillary thyroid carcinoma group than in the control group (33.3% vs. 11.4%, *p* < 0.05) [10]. Possible causes for the association between thyroid cancer and osteoporosis are listed below. First, TSH suppression therapy and hyperthyroidism may be risk factors for osteoporosis, as these factors enhance osteoclastic bone resorption in adults [19]. Levothyroxine replacement has been prescribed after total thyroidectomy to suppress TSH and supply thyroid hormones. TSH is suggested to be a key negative regulator of bone turnover. When it was suppressed below 0.4 mIU/L, the risk of osteoporosis increases in differentiated thyroid cancer patients [20]. TSH suppression and an older age have a synergistic effect on the reduction of bone marrow density [21]. Second, vitamin D deficiency may be a common risk factor for thyroid cancer and osteoporosis [22]. The major functions of vitamin D are to maintain homeostasis regarding the levels of calcium and phosphorus in the body and to preserve bone health. Moreover, vitamin D deficiency has been reported to be positively associated with thyroid cancer through the increased expression of p27, known as a tumor suppressor protein, by binding to the vitamin D receptor [23,24,25]. Third, autoimmunity may link thyroid cancer and osteoporosis by activating bone resorption. PTC patients with severe osteoporosis have a higher prevalence of TPO-Ab than patients without severe osteoporosis [10]. The presence of TPO-Ab is associated with osteoporotic fractures in euthyroid postmenopausal women [26]. On the other hand, the risk of thyroid cancer is not related to TPO-Ab levels but is associated with Tg-Ab levels [27]. However, the associations between autoimmune diseases and thyroid cancer are still controversial. Fourth, calcitonin deficiency after total thyroidectomy is a possible cause of osteoporosis. Calcitonin is secreted in C cells of the thyroid gland and acts as a potent inhibitor of osteoclastic bone resorption to maintain stable serum calcium levels [28,29]. However, the physiological action of calcitonin on bone tissue is not fully understood, although this hormone has been approved for the treatment of osteoporosis for approximately 30 years [30]. Fifth, the incidence rates of both thyroid cancer and osteoporosis are much higher in women than in men [31,32]. Therefore, the levels of female sex hormones, such as estrogen, could be related to the etiology of these two diseases. However, additional studies are needed on this topic.

The OR of fractures history was not significantly higher than 1, comparing participants with thyroid cancer with the control group. Few studies on the association between thyroid cancer and osteoporotic fractures have been conducted. In a previous study, a history of thyroidectomy did not increase the overall fracture risk in 630 women with respect to the community incidence rate [13]. In a large-scale case-control US study (*n* = 10,370), thyroid cancer was not related to an increased risk of fracture [9]. In a large population-based Korean study (*n* = 185,956), both high and low doses of levothyroxine treatment were linked to a higher risk of fracture in post thyroidectomy patients [17]. However, the association between hypo- or hyperthyroidism and fracture history was not decisive. In another study, low-dose levothyroxine was not associated with fracture risk, and antithyroid drugs significantly reduced the risk of fractures [16]. Thyroid cancer patients may be more frequently followed and actively treated with osteoporosis medication than controls. However, this topic needs to be studied further.

In our subgroup analysis, the ORs for osteoporosis were significantly higher in women with thyroid cancer only than in women in the control group. The incidence rates of osteoporosis and thyroid cancer were much higher in women than in men. However, the risk of osteoporosis was higher in the postmenopausal stage due to estrogen deficiency, and thyroid cancer was more common in premenopausal women than in postmenopausal women [33,34]. Therefore, additional studies need to be conducted to determine the mechanism of the link between female hormone levels and these two diseases. The cause of the inconsistency in the results between the men and women could be related to the sex differences in thyroid cancer incidence. The number of males with thyroid cancer was too small in our study, although to the total participant number is considered large. In our study, the total number of thyroid cancer patients was 1349, and 1235 patients (91.5%) were women. Therefore, the number of men (*n* = 114) may have been too small to detect an association between thyroid cancer and osteoporosis. A previous large population-based cohort study in Taiwan showed similar results. Among 1426 thyroid cancer patients, the women (*n* = 1208) showed a significantly increased incidence of osteoporosis after thyroid operation compared with the controls. However, the men (*n* = 218) did not show statistically significantly different results compared with the controls [12]. Moreover, in a US veteran study including 8689 male patients, osteoporosis was more common in male thyroid cancer patients than in the controls [9].

There are several strengths of this study. First, we adjusted for the possible risk factors of thyroid cancer or osteoporosis to minimize the effects of confounding factors such as BMI, aging, diabetes mellitus, a history of thyroid disease, smoking and alcohol consumption. Second, the models were adjusted to account for a history of arthritis because autoantibodies against RA, systemic inflammatory conditions and glucocorticoid use can increase the prevalence of osteoporosis [35]. Third, this was a large-scale population-based prospective study. There are several limitations. First, recall bias may have been present in this study because information about previous diseases and possible risk factors for osteoporosis were reported by the patients rather than retrieved from medical records. Second, osteoporosis was not diagnosed using bone mineral density measurements, as recommended by the WHO. Third, this database does not contain medication information, and medications such as steroids, diuretics and thyroxine can influence the incidence of osteoporosis. Fourth, the results of thyroid function tests, such as TSH, free T4, TPO-Ab and Ag-Ab tests, were not available, and thyroid function may influence the prevalence of osteoporosis. Fifth, specific thyroid cancer treatments were not investigated in this study, such as thyroid operations and RAI treatment. Sixth, this study has limitations as a cross-sectional study because we did not have the information as to when participants were diagnosed with thyroid cancer and fractures or osteoporosis. Seventh, we cannot evaluate according to thyroid cancer type, grade and stage due to lack of information. However, papillary thyroid carcinoma accounted for 97.2% of Korean thyroid cancer according to the previous national epidemiologic study [36].

## 5. Conclusions

Our study provides very important evidence of the relationship between thyroid cancer and the risk of osteoporosis. The adjusted OR of osteoporosis was significantly higher than 1, comparing the thyroid cancer group with the control group, especially in the female group. The adjusted OR of fractures was not significantly higher than 1, comparing the thyroid cancer group with the control groups. Additional studies are needed to determine the effect of thyroid-stimulating hormones, including free T4, TPO-Ab, Ag-Ab and estrogen.

## Figures and Tables

**Figure 1 ijerph-18-04732-f001:**
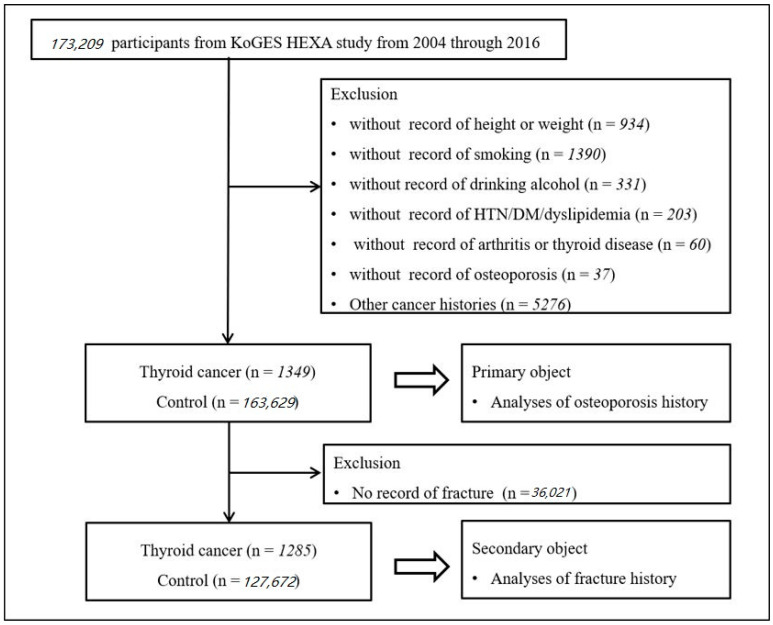
A schematic illustration of the participant selection process that was used in the present study. Of a total of 173,209 participants, 1349 thyroid cancer and 163,629 control (non-thyroid cancer) participants were selected.

**Table 1 ijerph-18-04732-t001:** General characteristics of participants.

Characteristics	Thyroid Cancer Histories		*p*-Value
	Yes	No	
Number (*n*, %)	1349 (100.0)	163,629 (100.0)	
Age (y, mean, SD)	52.3 (7.3)	53.0 (8.4)	<0.001 *
Sex (women, *n*, %)	1235 (91.5)	106,993 (65.4)	<0.001 *
BMI (kg/m^2^, mean, SD)	23.9 (2.9)	23.9 (2.9)	0.646
Hypertension (*n*, %)	270 (20.0)	31,706 (19.4)	0.555
Diabetes mellitus (*n*, %)	64 (4.7)	10,833 (6.6)	0.006 *
Dyslipidemia (*n*, %)	144 (10.7)	14,800 (9.0)	0.038 *
Arthritis (*n*, %)	124 (9.2)	15,621 (9.5)	0.659
Thyroid diseases (*n*, %)	253 (18.8)	7365 (4.5)	<0.001 *
Smoking (*n*, %)			<0.001 *
Non-smoker	1251 (92.7)	118,894 (72.7)	
Past smoker	61 (4.5)	23,973 (14.7)	
Current smoker	37 (2.7)	20,762 (12.7)	
Drinking alcohol (*n*, %)			<0.001 *
Non-drinker	901 (66.8)	82,443 (50.4)	
Past drinker	41 (3.0)	6237 (3.8)	
Current drinker	407 (30.2)	74,949 (45.8)	
Osteoporosis (*n*, %)	149 (11.0)	10,939 (6.7)	<0.001 *

* Independent *t*-test or Chi-square test, Statistical significance at *p* < 0.05.

**Table 2 ijerph-18-04732-t002:** Odds ratios of osteoporosis comparing participants with thyroid cancer to control.

Variables		Osteoporosis/Participants (*n*, %)	ORs for Osteoporosis			
			Crude	*p*-Value	Adjusted 1 ^†^	*p*-Value
Total participants (*n* = 164,978)						
	Thyroid cancer	149/1349 (11.0)	1.73 (1.46–2.06)	<0.001 *	1.41 (1.18–1.70)	<0.001 *
	Control	10,939/163,629 (6.7)	1		1	
Men (*n* = 56,750)						
	Thyroid cancer	1/114 (0.9)	1.05 (0.15–7.54)	0.961	0.98 (0.14–7.07)	0.982
	Control	473/56,636 (0.8)	1		1	
Women (*n* = 108,228)						
	Thyroid cancer	149/1235 (12.0)	1.26 (1.06–1.49)	0.010 *	1.43 (1.19–1.71)	<0.001 *
	Control	10,466/106,993 (9.8)	1		1	

* Logistic regression was analyzed, Significance at *p* < 0.05; ^†^ It was adjusted for age, sex, body mass index (BMI), hypertension, diabetes mellitus, dyslipidemia, arthritis, thyroid disease histories, smoking, and drinking alcohol.

**Table 3 ijerph-18-04732-t003:** Odds ratios of fracture histories comparing participants with thyroid cancer to control.

Variables		Fracture/Participants (*n*, %)	ORs for Fracture			
			Crude	*p*-Value	Adjusted 1 ^†^	*p*-Value
Total participants (*n* = 128,957)						
	Thyroid cancer	150/1285 (11.7)	0.92 (0.77–1.09)	0.307	1.00 (0.84–1.19)	0.982
	Control	16,119/127,672 (12.6)	1		1	
Men (*n* = 45,162)						
	Thyroid cancer	12/109 (11.0)	0.70 (0.39–1.28)	0.252	0.74 (0.41–1.35)	0.324
	Control	6734/45,053 (14.9)	1		1	
Women (*n* = 83,795)						
	Thyroid cancer	138/1176 (11.7)	1.04 (0.87–1.24)	0.687	1.06 (0.88–1.27)	0.528
	Control	9385/82,619 (11.4)	1		1	

^†^ It was adjusted for age, sex, body mass index (BMI), hypertension, diabetes mellitus, dyslipidemia, arthritis, thyroid disease histories, smoking, drinking alcohol, and osteoporosis.

## Data Availability

Release of the data by the authors is not legally allowed. All of the data are available on the database of the Korean genome and epidemiology study (KoGES) health examinee (HEXA) data in National Research Institute of Health (NIH) https://nih.go.kr/ (accessed on 3 November 2020). NIH permits access to all of these data via download for any researcher who promises to follow the research ethics.

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
