# Peer review of "Analyses of the Association between Thyroid Cancer and Osteoporosis/Fracture Histories: A Cross-Sectional Study Using KoGES HEXA Data"

_ijerph, 2021, doi:10.3390/ijerph18094732_

Round 1

Reviewer 1 Report

The Authors analyzed the association between thyroid cancer, osteoporosis and fracture history. The analysis was performed by comparing 1,349 participants with thyroid cancer and 163,629 control participants. They analyzed urban participants >= 40 years old, in South Korea. The Authors found that females with thyroid cancer were more likely to have osteoporosis than females without thyroid cancer (OR=1.43; p<0.001). However, the association between thyroid cancer and osteoporosis among males was not found. Moreover, the Authors did not find the association between fracture history and thyroid cancer.

The analysis was performed on a large group of participants, and the findings were consistent with literature. The paper is generally good-written, but it requires some corrections. Especially, wrong expressions were used describing odds ratios. So, language corrections performed by a statistician would enhance the paper.

Please, find detailed comments below.

General comments

  1. Section 2.3

What are the criterions for nondrinkers, past drinkers and current drinkers?

  1. It is not clear for me, what is the time relation between thyroid cancer and fractures or osteoporosis. Did the Authors consider all fractures through the whole life of the participants? For example, if a patient had a fracture at age of 10, and then had a thyroid cancer at age of 50, these two events are probably not associated with each other. Also, if the participant had osteoporosis 10 years prior the thyroid cancer, it is quite different situation compared to having osteoporosis 10 years after the thyroid cancer diagnosis. Did the Authors take it into account? If not, it should be clearly explained in methods section, that the whole history of thyroid cancer, osteoporosis or fractures was taken into account, regardless the time it occurred.
  2. Please consider adding numbers and percentages of osteoporosis patients to table 2 and 3. The table will be easier to interpret, as odds ratios alone are quite hard to interpret.
  3. In the discussion the Authors wrote: “However, the risk of osteoporosis was higher in the postmenopausal stage due to estrogen deficiency, and thyroid cancer was more common in premenopausal women than in post-menopausal women.” There is no such information in the results. There is also no citation for this statements. Please, give the source of this statement.

 Language comments

  1. Section 2.2

Authors wrote that the control group included “nonthyroid cancer” participants. It suggests that the control group included patients with cancer other than thyroid, which is misleading. Please, make a correction.

  1. Describing odds ratios.
    • The Authors wrote that “the adjusted ORs of thyroid cancer for fracture history were not statistically associated …”. It is wrong. It should be rather written that “the adjusted ORs were statistically larger than 1” or “the fracture occurrence was not statistically associated with thyroid cancer …”.
    • The Authors wrote “OR of thyroid cancer for osteoporosis”. I can infer from the context, that they wrote about OR of osteoporosis comparing participants with thyroid cancer to participants without it. However, it may be also interpreted as OR of thyroid cancer comparing osteoporosis to non-osteoporosis participants. Please, write it in more understandable way.
    • The Authors also wrote that “OR for osteoporosis was significantly higher in the thyroid cancer group than in the control group”. As, the OR in control group is always 1, it should rather be written that “OR for osteoporosis was significantly higher than 1”
    • The sentence should be corrected: “The odds ratios (ORs) of osteoporosis and fractures in patients with thyroid cancer were compared with those in controls”. The odds ratio itself compares these two groups.
    • The sentence should be corrected: “The OR of thyroid cancer with a history of fractures did not significantly differ compared with the control group.”
    • The sentence should be corrected: “In our subgroup analysis, the ORs for osteoporosis were significantly higher in the women with thyroid cancer only than in the control group”. It should be: “than in women in the control group”.
    • Please, correct sentences describing odds ratios, in the whole paper. For example, it can be described with following sentences:
      • “Osteoporosis was significantly associated with thyroid cancer”
      • “Participants with thyroid cancer are more likely to have osteoporosis then participants without thyroid cancer.
      • “Participants with thyroid cancer history had a higher odds of having osteoporosis than did control participants.”
  1. Please, correct the misspellings on page 6, line 3. There is word “though”, and it should rather be “through”.

Author Response

The analysis was performed on a large group of participants, and the findings were consistent with literature. The paper is generally good-written, but it requires some corrections. Especially, wrong expressions were used describing odds ratios. So, language corrections performed by a statistician would enhance the paper.

Response: Thank you for your comments. We edited sentence as below following your recommendation on page 3, line 27, 28, 32,33, 38-40.

3.1.1. General characteristics of participants

The proportions of women (91.5% vs 65.4%), patients with thyroid disease (18.8% vs 4.5%), nonsmokers (92.7% vs 72.7%), nondrinkers (66.8% vs 50.4%) and patients with osteoporosis (11.0% vs 6.7%) were higher in the thyroid cancer group than in the control group with statistical significance… The mean age was lower in the thyroid cancer group compared to the control group with statistical significance (Table 1)

3.1.2. ORs of thyroid cancer for osteoporosis

The adjusted ORs of thyroid cancer for osteoporosis were 1.41 (95% CI = 1.18–1.70, P <0.001) in all participants and 1.43 (95% CI = 1.19-1.71, P <0.001) in women (Table 2).

Point 1: What are the criterions for nondrinkers, past drinkers and current drinkers?

Response 1: Thank you for your comments. We changed following your recommendation on page 3, line 6,7

2.3. Survey

According to their alcohol consumption habits, the patients were categorized as nondrinkers, past drinkers, or current drinkers.

According to their alcohol consumption habits, the patients were categorized as nondrinkers (no history of alcohol consumption in their entire life), past drinkers (quit alcohol consumption), or current drinkers.

Point 2: It is not clear for me, what is the time relation between thyroid cancer and fractures or osteoporosis. Did the Authors consider all fractures through the whole life of the participants? For example, if a patient had a fracture at age of 10, and then had a thyroid cancer at age of 50, these two events are probably not associated with each other. Also, if the participant had osteoporosis 10 years prior the thyroid cancer, it is quite different situation compared to having osteoporosis 10 years after the thyroid cancer diagnosis. Did the Authors take it into account? If not, it should be clearly explained in methods section, that the whole history of thyroid cancer, osteoporosis or fractures was taken into account, regardless the time it occurred.

Response 2: Thank you for your comments. Unfortunately, we do not have the information at that times when participants diagnosed with thyroid cancer and fractures or osteoporosis. This is the limitation of cross-sectional study like ours. Therefore, we described about this point in the limitation section on page 7, line 7-9. 

“Sixth, this study has limitation as a cross-sectional study because we did not have the information when participants diagnosed with thyroid cancer and fractures or osteoporosis.”

Point 3: Please consider adding numbers and percentages of osteoporosis patients to table 2 and 3. The table will be easier to interpret, as odds ratios alone are quite hard to interpret.

Response 3: Thank you for your comments. We added numbers and percentage of osteoporosis patients to table 2 and 3 on page 5.

Point 4: In the discussion the Authors wrote: “However, the risk of osteoporosis was higher in the postmenopausal stage due to estrogen deficiency, and thyroid cancer was more common in premenopausal women than in post-menopausal women.” There is no such information in the results. There is also no citation for this statements. Please, give the source of this statement.

Response 4: Thank you for your comments. We added citations for these statements on page 6, line 31.

“However, the risk of osteoporosis was higher in the postmenopausal stage due to estrogen deficiency, and thyroid cancer was more common in premenopausal women than in postmenopausal women [33,34].”

 Language comments

Point 5: Authors wrote that the control group included “nonthyroid cancer” participants. It suggests that the control group included patients with cancer other than thyroid, which is misleading. Please, make a correction.

Response 5: Thank you for your comments. We changed from nonthyroid cancer to no history of thyroid cancer on page 2, line 40.

2.2. Participants Selection

“Finally, 1,349 thyroid cancer and 163,629 control (no history of thyroid cancer) participants were selected (Fig. 1)”

Point 6 : Describing odds ratios.

The Authors wrote that “the adjusted ORs of thyroid cancer for fracture history were not statistically associated …”. It is wrong. It should be rather written that “the adjusted ORs were statistically larger than 1” or “the fracture occurrence was not statistically associated with thyroid cancer …”.

Response 6: Thank you for comment. We change that sentence like below on page 3, result section.

3.1.3. ORs of thyroid cancer for fracture histories

Response: The adjusted ORs of thyroid cancer for fracture history were not statistic significantly larger than 1 in the entire group of participants, men only or women only (Table 3).

The Authors wrote “OR of thyroid cancer for osteoporosis”. I can infer from the context, that they wrote about OR of osteoporosis comparing participants with thyroid cancer to participants without it. However, it may be also interpreted as OR of thyroid cancer comparing osteoporosis to non-osteoporosis participants. Please, write it in more understandable way.

Response: Thank you for comment. We changed that sentence into more understandable way on page 3 and table 2 and 3.

“OR of thyroid cancer for osteoporosis” was changed into “ORs of osteoporosis comparing participants with thyroid cancer to control”

The Authors also wrote that “OR for osteoporosis was significantly higher in the thyroid cancer group than in the control group”. As, the OR in control group is always 1, it should rather be written that “OR for osteoporosis was significantly higher than 1”

Response: Thank you for your comment. We changed that sentence like below on page 6, line 7,8.

“The ORs for osteoporosis were significantly higher in the thyroid cancer group than in the control group.”

“The ORs of osteoporosis was significantly higher than 1 comparing participants with thyroid cancer to control.”

The sentence should be corrected: “The odds ratios (ORs) of osteoporosis and fractures in patients with thyroid cancer were compared with those in controls”. The odds ratio itself compares these two groups.

Response: We changed that sentence as below on page 2, line 21, 22.

“The odds ration itself compares these two groups.”

The sentence should be corrected: “The OR of thyroid cancer with a history of fractures did not significantly differ compared with the control group.”

Response: We changed that sentence as below on page 6, line 12,13.

“The OR of fractures history was not significantly higher than 1 comparing participants with thyroid cancer to the control group.”

The sentence should be corrected: “In our subgroup analysis, the ORs for osteoporosis were significantly higher in the women with thyroid cancer only than in the control group”. It should be: “than in women in the control group”.

Response: We changed that sentence as below on page 7, line 26, 27.

“The ORs for osteoporosis were significantly higher in the women with thyroid cancer only than in women in the control group.”

Please, correct sentences describing odds ratios, in the whole paper. For example, it can be described with following sentences:

“Osteoporosis was significantly associated with thyroid cancer”

“Participants with thyroid cancer are more likely to have osteoporosis then participants without thyroid cancer.

“Participants with thyroid cancer history had a higher odds of having osteoporosis than did control participants.”

Response: Thank you for comments. According to your recommendation, we changed conclusion as below on page 7, line 18-21

“The adjusted OR of osteoporosis was significantly higher than 1 comparing thyroid cancer group to the control group, especially in the female group. The adjusted OR of fractures was not significantly higher than 1 comparing the thyroid cancer group to the control groups.”

Point 7: Please, correct the misspellings on page 6, line 3. There is word “though”, and it should rather be “through”.

Response: “though” was changed into “though” on page 6, line 32.

Reviewer 2 Report

The authors present an epidemiological study on thyroid cancer, osteoporosis and related fractures. The study is based on robust data collected from 2004 to 2016 within the Korean Genome and Epidemiology Study from where particular informations were retrieved. In the present analysis, a total of 1,349 participating subjects was diagnosed with thyroid cancer and 163,629 control participants were cancer free. The results received by using a logistic regression model are as follows: The adjusted ORs of thyroid cancer for osteoporosis were 1.41 (95% CI = 1.18–1.70, P <0.001) for all participants and 1.43 (95% CI = 1.19-1.71, P <0.001) in the women. The adjusted ORs of thyroid cancer for fracture history were not significantly associated within the entire group of participants or separatly for men only or women only. Based on the results, the authors concluded that the adjusted OR for osteoporosis was significantly higher in the thyroid cancer group than in the control group, especially in the female group. On the other hand, the adjusted OR for fracture history did not significantly differ.

Minor comments: As thyroid cancer is quite heterogenous group,it would be useful to state what kind of cancer was represented in the study according to cancer type, grade and stage.

In Korea, screening for thyroid cancer was performed. Can author discuss if this is a bias for the presented data?

Author Response

Point 1: As thyroid cancer is quite heterogenous group,it would be useful to state what kind of cancer was represented in the study according to cancer type, grade and stage.

Response 1: Thank you for your comment. Our data base does not contain information for the thyroid cancer type, grade and stage. Therefore, we added these points in the limitation section on page 7, line 9-12

“Seventh, we cannot evaluate according to thyroid cancer type, grade and stage due to lack of information. However, papillary thyroid carcinoma accounted for 97.2% of Korean thyroid cancer according to the previous national epidemiologic study [36].”

Point 2: In Korea, screening for thyroid cancer was performed. Can author discuss if this is a bias for the presented data?

Response 2: Thyroid cancer is 5th most common female cancer in US and 2nd most common female cancer in Korea. Although the etiology for thyroid cancer were diverse, early thyroid evaluation could be one of the reasons for high incidence rate in Korea. However, there are many other studies to support a positive association between thyroid cancer and osteoporosis in other country[1-3]. 

  1. Lin, S.-Y.; Lin, C.-L.; Chen, H.-T.; Kao, C.-H. Risk of osteoporosis in thyroid cancer patients using levothyroxine: a population-based study. Current medical research and opinion 2018, 34, 805-812.
  2. Notsu, M.; Yamauchi, M.; Morita, M.; Nawata, K.; Sugimoto, T. Papillary thyroid carcinoma is a risk factor for severe osteoporosis. Journal of bone and mineral metabolism 2020, 38, 264-270.
  3. Hung, C.-L.; Yeh, C.-C.; Sung, P.-S.; Hung, C.-J.; Muo, C.-H.; Sung, F.-C.; Jou, I.-M.; Tsai, K.-J. Is partial or total thyroidectomy associated with risk of long-term osteoporosis: a nationwide population-based study. World journal of surgery 2018, 42, 2864-2871.